# Differences in Pedestrian Behavior at Crosswalk between Communicating with Conventional Vehicle and Automated Vehicle in Real Traffic Environment

Masahiro Taima * and Tatsuru Daimon

Faculty of Science and Technology, Keio University, 3-14-1 Hiyoshi, Kohoku-ku, Yokohama 223-8522, Japan
* Correspondence: taima@keio.jp

**Abstract:** In this study, we examine the differences in pedestrian behavior at crosswalks between communicating with conventional vehicles (CVs) and automated vehicles (AVs). To analyze pedestrian behavior statistically, we record the pedestrian's position (x- and y-coordinates) every 0.5 s and perform a hot spot analysis. A Toyota Prius (ZVW30) is used as the CV and AV, and the vehicle behavior is controlled using the Wizard of Oz method. An experiment is conducted on a public road in Odaiba, Tokyo, Japan, where 38 participants are recruited for each experiment involving a CV and an AV. The participants cross the road after communicating with the CV or AV. The results show that the pedestrians can cross earlier when communicating with the CV as compared with the AV. The hot spot analysis shows that pedestrians who communicate with the CV decide to cross the road before the CV stops; however, pedestrians who communicate with the AVs decide to cross the road after the AV stops. It is discovered that perceived safety does not significantly affect pedestrian behavior; therefore, earlier perceived safety by drivers' communication and external human–machine interface is more important than higher perceived safety for achieving efficient communication.

**Keywords:** automated vehicle (AV); pedestrians; Wizard of Oz method; hot spot analysis; external human–machine interface (eHMI)



## 1. Introduction

Communication design is important when implementing automated vehicles (AVs) in a real traffic environment because smooth communication between AVs and road users (e.g., pedestrians and drivers) is necessary to ensure a safe and efficient traffic environment [1–3]. In fact, several traffic accidents and studies indicate that insufficient communication between AVs and road users can result in unsafe and inefficient scenarios. Wang and Li [4] and Petrović, Mijailović, and Pešić [5] analyzed AV crash records, such as California's Report of Traffic Collision Involving an Autonomous Vehicle Database (Californian Department of Motor Vehicles, 2018), and discovered that some fatal crashes between AVs and conventional vehicles (CVs) occurred when the CV changed lanes while closely trailing an AV. Communication between an AV and a CV was not the only reason that caused these serious accidents; however, safe and efficient communications can reduce such accidents. Moreover, several experiments showed that implicit and explicit communication (vehicle behavior and external human–machine interface (HMI)) increased a road user's sense of safety and reduced their decision time [6–11]. Moreover, the safety of other road users [12–15] and traffic efficiency [16] must be guaranteed after introducing AVs into the traffic environment. Therefore, communication design is important when introducing AVs in a real traffic environment.

To achieve an appropriate communication design between an AV and road users, differences in road user responses when communicating with CVs and AVs must be investigated. Previous research indicated differences in the decisions (e.g., pedestrian decided to cross a crosswalk earlier in cases involving a CV instead of an AV) and perceived

safety (e.g., pedestrian perceived high level of safety in cases involving a CV instead of an AV) of road users [6–8,10,17,18]. Appropriate communication designs must be designed to reduce unsafe and inefficient communication caused by differences in road user responses between CVs and AVs.

Unlike VR experiments [19,20], in real road experiments, road user's behavioral aspects are focused on less. The effects of differences between CVs and AVs on road users' psychology (decision and perceived safety) and behavior (road user's location and moving speed) have been analyzed in previous studies. Regarding public road experiments, few studies have focused on behavioral aspects, as it is difficult to monitor the road user's location (x- and y-coordinates of the road). Therefore, the effects of the difference between communication with CVs and AVs on the behavior of road users remain ambiguous.

This study focuses on communication between AVs and pedestrians at crosswalks; this is because the safety of pedestrians must be prioritized instead of the safety of other road users, since serious traffic accidents have occurred during pedestrian crossing [18,21–23].

In addition, this study focuses on the effect of drivers' communication (eye contact and gesture) on pedestrian behavior [24–26]. It is important to focus on the importance of the external human–machine interface (eHMI). Previous research shows that several explicit and implicit cues affect pedestrian behavior. As an explicit cue, vehicle behavior (speed and acceleration) affects pedestrian behavior, and several studies indicated that vehicle behavior was the most important cue for pedestrians [11,27–29]. As implicit cues, drivers' communication and eHMI affects pedestrian behavior as well as increases pedestrians' perceived safety and certainty of decision [6–8,10,17,18]. In this study, we focus on the driver's communication.

To investigate the effect of no driver communication on pedestrian behavior, we form the following hypothesis:

**Hypothesis 1 (H1).** *A pedestrian crosses earlier after communicating with a CV instead of an AV.*

In addition, high perceived safety may affect pedestrian behavior [30,31]; therefore, we form the following hypothesis:

**Hypothesis 2 (H2).** *High perceived safety causes pedestrian to cross earlier after communicating with an AV.*

In the following sections, firstly we describe the experiment and conditions in Section 2, then the analysis in Section 3. The results are discussed based on the hypotheses in Section 4.

## 2. Materials and Methods

### 2.1. Experiment Location and Period

We conducted an experiment in Odaiba, Tokyo, from November 28 to 29, 2018. In the experiment, CVs and AVs traversed on the road. We observed the communication between the participant and vehicle (CV or AV) at the crosswalk (Figure 1).

### 2.2. Vehicle and Devise

A Toyota Prius (ZVW30) was used as the CV and AV (Figure 2). For the case involving the AV, a sticker was attached to the vehicle to display the message "in automatic mode," and a laser range finder was attached to the roof. We used the Wizard of Oz method [29,32] to conceal the driver to mimic a driverless scenario (Figure 2). The Wizard of Oz experiment method has been used within the field of robot–human interaction for a long period. In this method, the human is simulated to test robot behaviors. In the present study, a human hid in the driver's seat, and manipulated the vehicle. The AV and the CV were controlled to have the same vehicle behavior by automatic running.

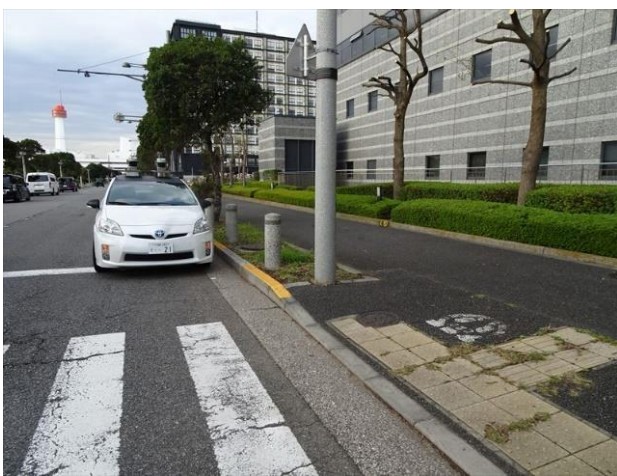

**Figure 1.** Study area. Participants communicated with vehicle and began crossing from right side of image.

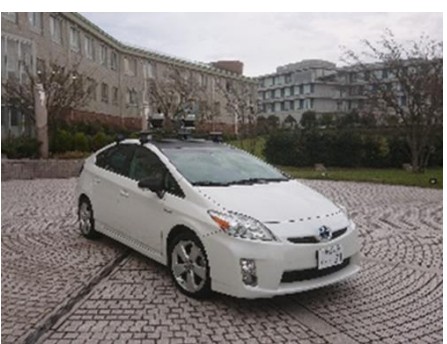 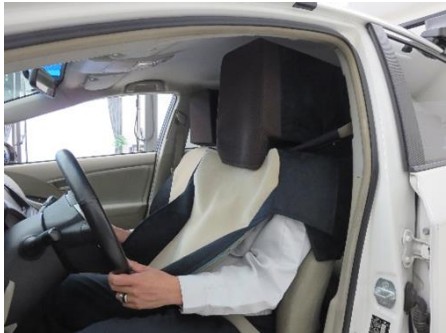

**Figure 2.** Vehicle and Wizard of Oz method.

To observe the positions of the pedestrians and vehicles, a camera was attached to the pole. We used three cameras in the experiment (see Figure 3).

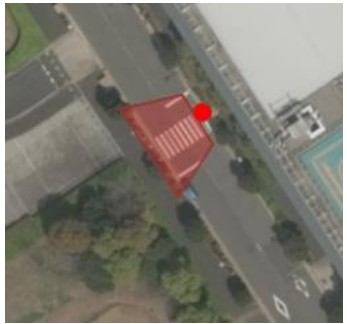 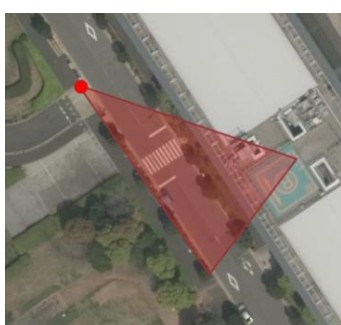 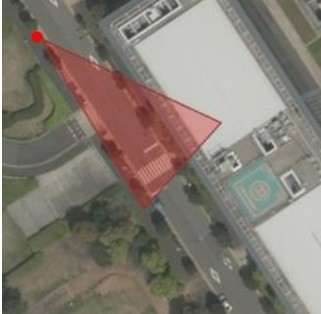

**Figure 3.** Three camera locations.

### 2.3. Positions of Pedestrian and Vehicle

Using video data, the positions of the pedestrians and vehicles were recorded. We set a corner of the crosswalk as the origin (x = 0 m, y = 0 m). We recorded the coordinates of the participant's right foot and the vehicle's right tire every 0.5 s. Data about positions of vehicles and pedestrians were collected both automatically (radar attached on vehicle) and manually (Observers confirmed the positions by video).

### 2.4. Scenario

Two scenarios (CV or AV) were implemented in the experiment (Table 1).

**Table 1.** Conditions of Scenario 1 and 2.

| Scenario | Vehicle | Driver | Passenger |
|---|---|---|---|
| Scenario 1 | CV | Yes | No |
| Scenario 2 | AV | No (Wizard of Oz) | No |

*2.5. Procedure*

The experiment was conducted as follows:

- Participants were informed that the AV will not stop in front of the crosswalk in some cases.
- The vehicle (CV or AV) waited at a location 60 m from crosswalk.
- The participant stood in front of the crosswalk (pedestrian could not watch the vehicle start, because the instructor controlled the pedestrian's body direction).
- The instructor instructed the participant to cross the road alone (each participant crossed the road once).
- Before the pedestrian crossed the road, the vehicle (CV or AV) approached the crosswalk at approximately 20 km/h and stopped in front of the crosswalk.
- The pedestrian communicated with vehicle (CV or AV) and then crossed the road (behavior was main communication tool by both pedestrians and vehicles).
- After the experiment, the participant provided answers to a questionnaire and an interview.

We conducted a between-subjects experiment. We used a temporary employment agency to recruit participants. We used similar sample sizes for each age group (e.g., 20–29 years old) in Scenarios 1 and 2. Hence, 38 participants were recruited for each of the Scenarios 1 and 2 (Table 2).

**Table 2.** Sample size in scenario 1 and 2.

| Age | Scenario 1 | Scenario 2 |
|---|---|---|
| 20–29 | 16 | 16 |
| 30–39 | 5 | 4 |
| 40–49 | 4 | 0 |
| 50–59 | 2 | 4 |
| 60–69 | 6 | 9 |
| 70–79 | 4 | 5 |
| 80–89 | 1 | 0 |
| Total | 38 | 38 |

*2.6. Questionnaire and Interview*

After performing the experiment, the participants were presented with a question regarding their perceived safety when crossing, as follows: "Did you perceive safety?" (regarding the psychological aspect, we not only asked "Did you perceive safety?", but also "confidence in communication" and "Feeling to be yielded by vehicle". In the statistical test, these questions were highly correlated, therefore, we only used the question "Did you perceive safety?" in our analysis). The participants responded on a seven-point Likert scale (1 = strongly disagree, 2 = disagree, 3 = slightly disagree, 4 = neutral, 5 = slightly agree, 6 = agree, 7 = strongly agree). Moreover, the interviewer asked the participants to elaborate on their answer.

*2.7. Analysis*

To examine the effect of vehicle type (CV or AV) on pedestrian behavior (H1), we compared the movements of two groups: participants in Scenario 1 (Group 1) and Scenario 2 (Group 2).

The time at which the vehicle stopped at the stop line in front of the crosswalk was defined as 0 s. We analyzed the pedestrian's movement from $-7.0$ to $3.0$ s (during this period, most pedestrians had finished crossing). The average position of the pedestrians at each time point is defined as follows:

$$x_{g_i t_j} = \frac{\sum_{k=1}^{n_{g_i}} x_{g_i t_j p_k}}{n_{g_i}} \tag{1}$$

$$y_{g_i t_j} = \frac{\sum_{k=1}^{n_{g_i}} y_{g_i t_j p_k}}{n_{g_i}} \tag{2}$$

$x_{g_i t_j}$ : Pedestrian's $x -$ coordinate at time $j$ in group $i$
$y_{g_i t_j}$ : Pedestrian's $y -$ coordinate at time $j$ in group $i$
$p_k$ : Participant $k$
$i = 1, 2$
$j = -7.0, -6.5, \ldots 2.5, 3.0$
$k = 1, 2, \ldots, n_{g_i}$

The average trajectories of the pedestrian's movement were constructed by connecting each average position.

To examine the differences in pedestrian movement in each group, a hot spot analysis [33,34] was performed. The hot spot analysis can indicate the location with spatially clustered high and low values. First, we constructed a 0.5 m × 0.5 m mesh on the crosswalk. Second, we counted the number of participants for each mesh. Third, we corrected the number of participants in each mesh as the Getis-Ord $G_i*$ based on the relationship with the adjacent mesh.

$$G_a* = \frac{\sum_{b=1}^{n} w_{ab} p_b - \overline{P} \sum_{b=1}^{n} w_{ab}}{S \sqrt{\frac{\left[ n \sum_{b=1}^{n} w_{ab}^2 - \left( \sum_{b=1}^{n} w_{ab} \right)^2 \right]}{n-1}}} \tag{3}$$

$$\overline{P} = \frac{\sum_{b=1}^{n} p_b}{n} \tag{4}$$

$$S = \sqrt{\frac{\sum_{b=1}^{n} p_b^2}{n} - \left( \overline{P} \right)^2} \tag{5}$$

$w_{ab}$ : Spatial weight between meshes $a$ and $b$
$p_b$ : Number of participants
$n$ : Number of mesh $b$

We statistically analyzed the cluster of pedestrian positions using the *p*-value of $G_i*$. By performing a hot spot analysis, we analyzed the differences in pedestrian movement in each group.

Moreover, we analyzed the differences in pedestrian movement speed in each group. We constructed a box plot every 0.5 s and conducted the Student's *t*-test between groups. The pedestrian's movement speed $v$ is defined as follows:

$$v_{g_i t_j} = \frac{\sum_{k=1}^{n_{g_i}} \frac{\left| \sqrt{\left( x_{g_i t_j p_k} - x_{g_i t_{j-1} p_k} \right)^2 + \left( y_{g_i t_j p_k} - y_{g_i t_{j-1} p_k} \right)^2} \right|}{t_j - t_{j-1}}}{n_{g_i}} \tag{6}$$

$v_{g_i t_j}$ : Pedestrian's movement speed at time $j$ in group $i$
$x_{g_i t_j p_k}$ : Pedestrian $k'$s $x -$ coordinate at time $j$ in group $i$
$y_{g_i t_j p_k}$ : Pedestrian $k'$s $y -$ coordinate at time $j$ in group $i$
$i = 1, 2$
$j = -7.0, -6.5, \ldots 2.5, 3.0$

$k = 1, 2, \ldots, n_{g_i}$

To examine the effect of perceived safety on pedestrian behavior (H2), we compared the movements of two groups: the participants who perceived safety (participants who provided a score exceeding 5 for the perceived safety question; Group 2a) and those who did not perceive safety (participants who provided a score of less than 3 for the perceived safety question, Group 2b).

We analyzed the differences between groups using similar methods in Section 2, i.e., the average trajectory, hot spot, and movement speed for each group.

### 3. Results

#### 3.1. Effect of Vehicle Type (CV or AV) on Pedestrian Behavior

Figure 4 shows the average positions of the pedestrian. Figure 5 shows the pedestrian's position at −7.0 and 0 s in each group, i.e., Groups 1 and 2, based on hot spot analysis. Comparing Group 1 (N = 38) and Group 2 (N = 38), the participants of Group 1 walked faster and began crossing before the vehicle (CV) stopped (see Figure 5a,c). By contrast, Group 2 did not begin crossing before the vehicle (AV) stopped and remained in similar positions from −7.0 to 0 s (see Figures 4b and 5b,d). Figure 6 shows the pedestrian's movement speed. We observed that Group 2 maintained a low speed for a longer time than Group 1 (e.g., the speed of Group 1 was less than 0.3 m/s until −3.5 s, whereas that of Group 2 was less than 0.3 m/s until −1.5 s. At 0 s, the participants of Group 1 walked faster as compared with those of Group 2 ($p < 0.001$)).

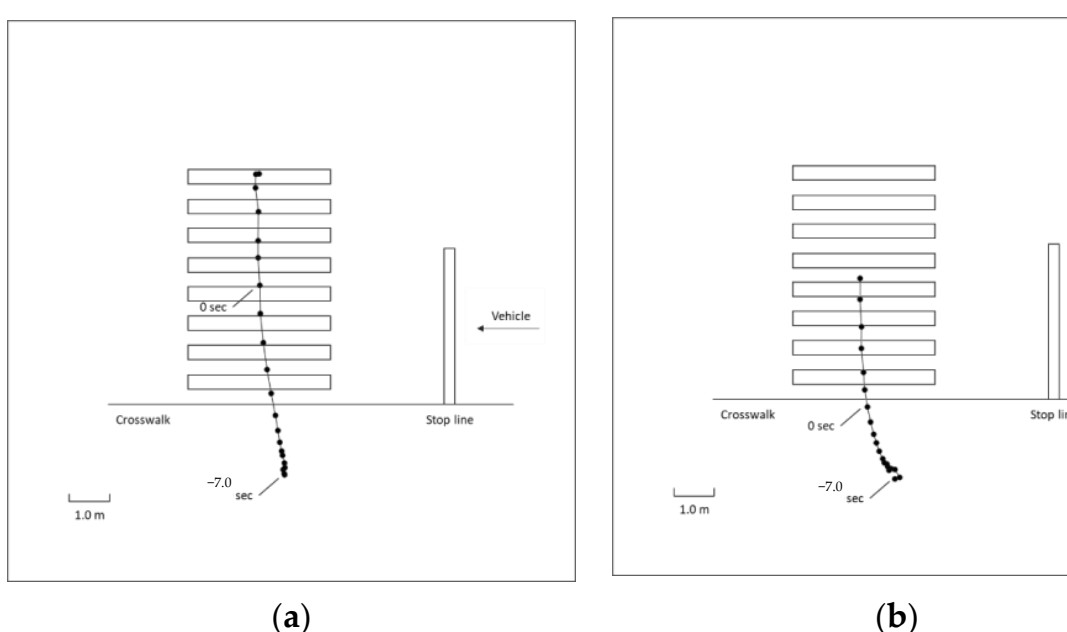

**(a)** **(b)**

**Figure 4.** Average positions of pedestrian in Group 1 (CV) and Group 2 (AV). (**a**) Group 1. (**b**) Group 2.

#### 3.2. Effects of Perceived Safety on Pedestrian Behavior

Figure 7 shows the average positions of the pedestrians. Figure 8 shows the pedestrian's position at −7.0 and 0 s in each group, i.e., Group 2a (N = 23, 5.59 (1.44)) and Group 2b (N = 13, 2.31(0.78)), based on hot spot analysis. It was discovered that both Group 2a and Group 2b did not begin crossing before the vehicle (AV) stopped, and most pedestrians remained in similar positions from −7.0 to 0 s (see Figure 7). However, some pedestrian crossed earlier in Group 2a (Figure 7a). Figure 9 shows the pedestrian's movement speed. We did not observe significant differences between Groups 2a and 2b.

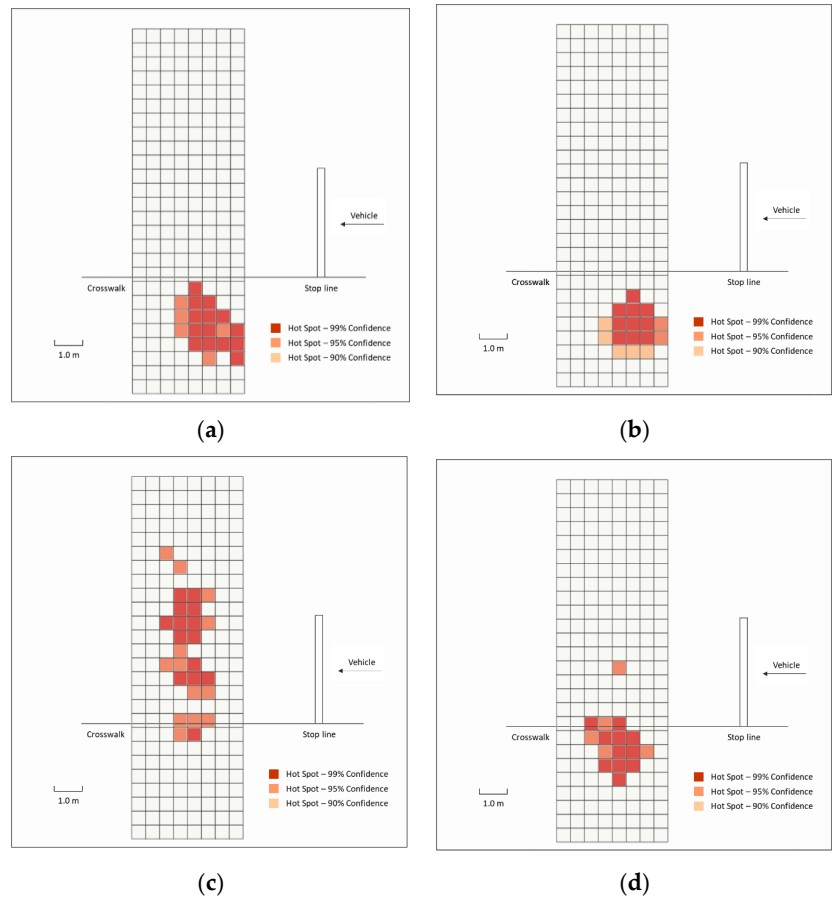

**Figure 5.** Hot spot analysis of pedestrian's position in Group 1 (CV) and Group 2 (AV). (**a**) Group 1 at −7.0 s. (**b**) Group 2 at −7.0 s. (**c**) Group 1 at 0 s. (**d**) Group 2 at 0 s.

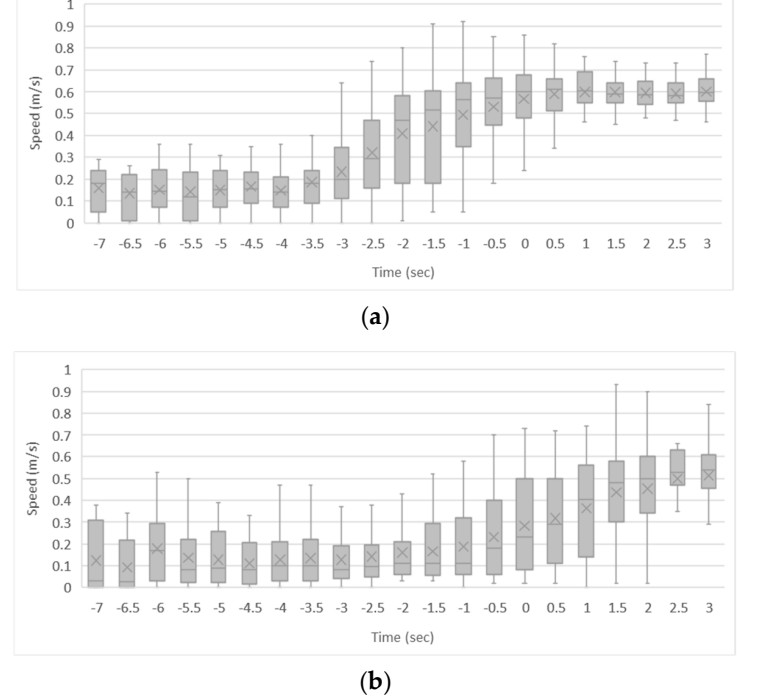

**Figure 6.** Pedestrian's movement speed in Group 1 (CV) and Group 2 (AV). (**a**) Group 1. (**b**) Group 2.

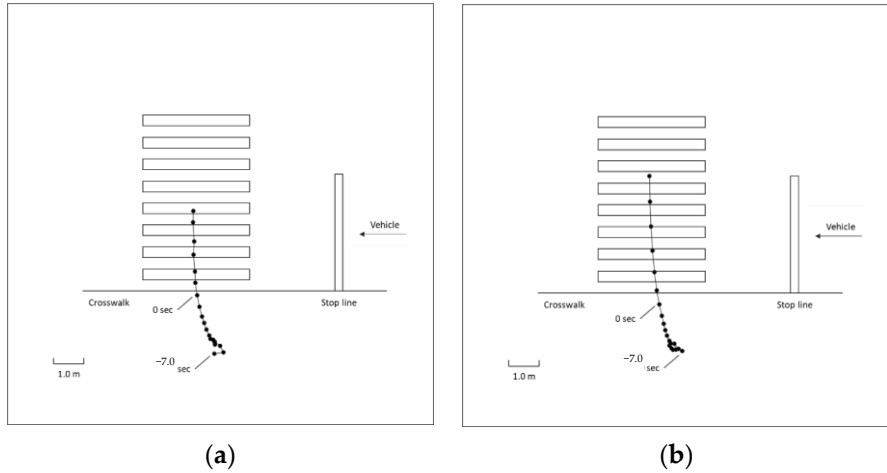

**Figure 7.** Average positions of pedestrian in Group 2a (high perceived safety) and Group 2b (low perceived safety). (**a**) Group 2a. (**b**) Group 2b.

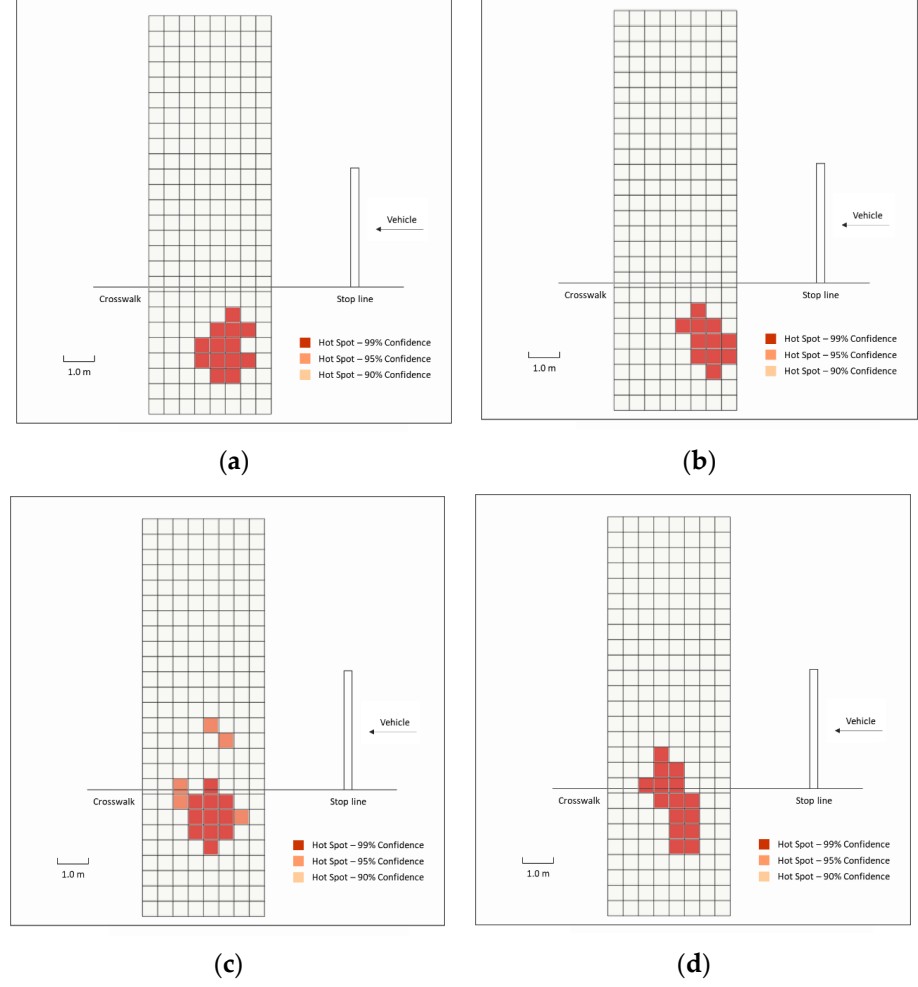

**Figure 8.** Hot spot analysis of pedestrian's position in Group 2a (high perceived safety) and Group 2b (low perceived safety). (**a**) Group 2a at −7.0 s. (**b**) Group 2b at −7.0 s. (**c**) Group 2a at 0 s. (**d**) Group 2b at 0 s.

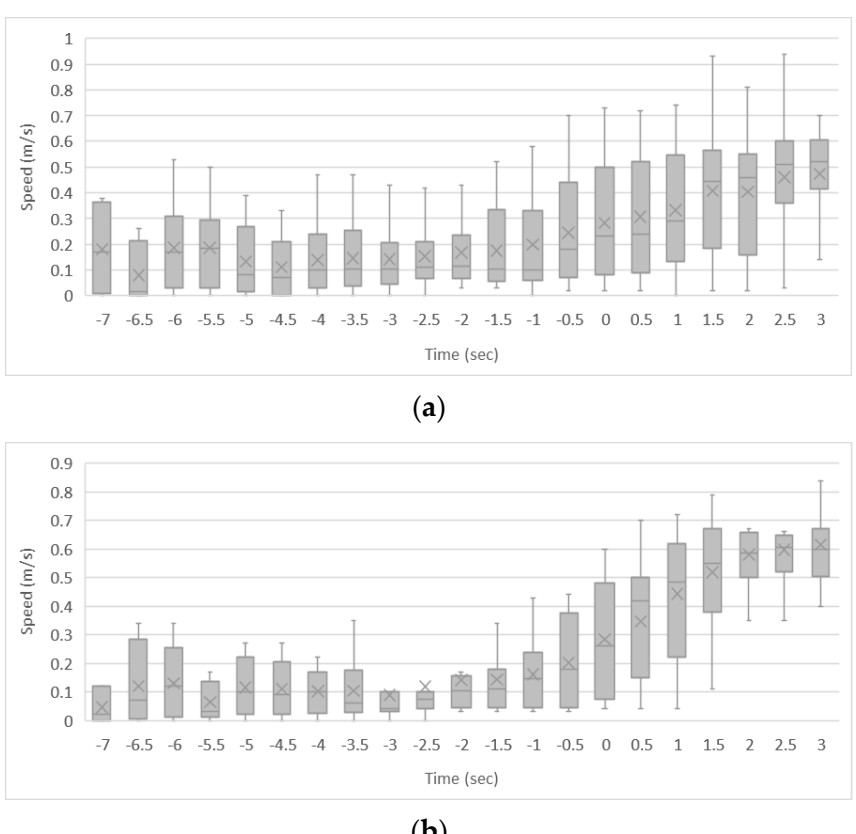

**Figure 9.** Pedestrian's movement speed in Group 2a (high perceived safety) and Group 2b (low perceived safety). (**a**) Group 2a. (**b**) Group 2b.

## 4. Discussion

Based on our experimental results, we discovered that the hypothesis "H1: Pedestrians crossed earlier when communicating with the CV instead of the AV" was correct. The participants in Group 1 began crossing before the CV stopped, and then walked 3 m on a crosswalk when the CV stopped (Figure 4a). By contrast, the participants in Group 2 hesitated to cross while the AV was moving and waited to cross until the AV stopped (Figure 4b). Figure 5 shows the significant difference between Groups 1 and 2.

The questionnaire and interview indicated that perceived safety and drivers contributed to the difference. Group 1 perceived higher safety than Group 2 ($p < 0.05$). The average score was 5.77 in Group 1, and 5.00 in Group 2. The perceived safety caused the participants to cross earlier, and several studies indicated the same results [6,21,35]. In the interview, the participants provided their reason for crossing the road. Most participants in Group 2 decided to cross based on the vehicle behavior (e.g., "I decided to cross when the vehicle was about to stop"). However, the participants in Group 1 decided to cross based on not only on vehicle behavior, but also communication with the driver (eye contact and gesture). In particular, 60.5% of the participants (23/38) in Group 1 answered that they decided to cross based on the driver's eye contact and gestures. Previous studies show that vehicle behavior is the most important cue for communication [21,29,36], and that pedestrians can decide to cross only based on vehicle behavior. However, this study showed that many pedestrians depended on communication with the driver, which enabled them to cross earlier. This result supports the importance of eHMI in efficient communication with AVs.

We could not validate the hypothesis "H2: High perceived safety causes pedestrians to cross earlier after communicating with AV.". We did not observe a significant difference in the participants' behavior between Group 2a (high perceived safety) and Group 2b (low perceived safety). The perceived safety between Groups 2a and 2b indicated a

significant difference ($p < 0.001$). The average scores were 5.86 and 2.48 for Groups 2a and 2b, respectively. However, the participants' behavior did not differ significantly, and most participants in Groups 2a and 2b did not decide to cross before the AV stopped (Figures 7 and 8, respectively). Moreover, although the perceived safety in Group 2a (AV) was higher than that in Group 1 (CV), Group 1 decided to cross earlier.

These results indicate that earlier perceived safety is more important than higher perceived safety for achieving efficient communication. Previous studies show that perceived safety is vital to the crossing decision of pedestrians [6,21,30,31,35]. However, it was shown in this study that the effect of perceived safety on earlier crossing decisions was less prominent than that of drivers' communication. Group 2a indicated high perceived safety; however, the participants hesitated to cross until the AV stopped (Figures 7a, 8c and 9a) because the AV was driverless. The participants in Groups 2a and 2b decided to cross based on only the AV behavior; therefore, eHMI may enable pedestrians to cross earlier. Therefore, earlier perceived safety by the driver's communication and eHMI is more important than higher perceived safety for achieving efficient communication.

Some participants in Group 2a decided to cross earlier (Figure 8c). In the interview, one participant answered that he did not verify the environment because he realized that the surrounding was silent, which prompted him to cross earlier. In this case, the vehicle type (CV or AV) did not pose a significant effect; however, the AV must be aware of such types of pedestrians. Meanwhile, it appeared that some participants over-trusted the AV; they indicated that they expected the AV to stop while they were crossing.

This research did not directly contribute to policy makers or other stakeholders responsible for AV and road safety, however, will be basic information for several AV's communication design, such as vehicle behavior and external HMI for communication.

Some limitations exist in this study. Because of the limited funding and experimental period, we could not control the sample size of each age group. Therefore, the differences in age could not be examined [37,38]. As the number of elderly people is fewer than that of young people, the behavior of the elderly must be examined more comprehensively. In addition, the effect of driving experience must be considered [39]. Groups 2a and 2b had small sample sizes (N = 23 in Group 2a; N = 13 in Group 2b); therefore, the accuracy of analysis of these groups was lower than that of Groups 1 and 2. The study was performed on public roads. We controlled the environment for every experiment; however, public road users (pedestrians and vehicles) may affect the participant's decision. The same vehicle type was used as the CV and AV; therefore, we could not examine the effect of the vehicle appearance [21,37,40–42]. Furthermore, we could not examine the effect of the participants' knowledge regarding AVs. If the participant learned the AV's behavior and mechanism through education and repeated communication, he/she participant might decide to cross earlier. Hence, we should examine the effect of education and conduct a time-series study.

## 5. Conclusions

In this study, we examined the differences in pedestrian behavior when communicating with CVs and AVs. The results showed that the pedestrians decided to cross earlier after communicating with a CV than with an AV. We observed that pedestrians who communicated with a CV decided to cross before the CV stopped; however, pedestrians who communicated with an AV decided to cross after the AV stopped. These differences were caused primarily by driver communication (eye contact and gestures). Most pedestrians who communicated with the CV decided to cross based on the driver's eye contact and gestures. Pedestrians who communicated with the AVs decided to cross based on only the AV's behavior; therefore, eHMI, instead of the driver's communication, might be more effective for achieving efficient communication. Comparing pedestrians who had high perceived safety and low perceived safety in cases involving communication with the AV, it was discovered that perceived safety did not significantly affect pedestrian behavior. Therefore, earlier perceived safety is more important than perceived safety for achieving

efficient communication. Further research pertaining to differences in age groups, effects of eHMI, and effects of education (pedestrian's knowledge about AV) should be conducted.

**Author Contributions:** Conceptualization, M.T. and T.D.; methodology, M.T. and T.D.; software, M.T. and T.D.; validation, M.T. and T.D.; formal analysis, M.T. and T.D.; investigation, M.T. and T.D.; resources, T.D.; data curation, M.T. and T.D.; writing—original draft preparation, M.T.; writing—review and editing, M.T. and T.D.; visualization, M.T.; supervision, T.D.; project administration, T.D.; funding acquisition, T.D. All authors have read and agreed to the published version of the manuscript.

**Funding:** This study was supported by the Council for Science, Technology, and Innovation (CSTI), Cross-ministerial Strategic Innovation Promotion Program (SIP), and the "Large-scale Field Operational Test for Automated Driving Systems" (funding agency: NEDO).

**Institutional Review Board Statement:** The study was conducted in accordance with the Declaration of Helsinki, and approved by the Institutional Review Board (or Ethics Committee) of Keio University (protocol code 31–35 and 1 April 2019).

**Informed Consent Statement:** Informed consent was obtained from all subjects involved in the study.

**Data Availability Statement:** Not applicable.

**Conflicts of Interest:** The authors declare no conflict of interest.

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
