# Peer review of "Differences in Pedestrian Behavior at Crosswalk between Communicating with Conventional Vehicle and Automated Vehicle in Real Traffic Environment"

_safety, 2022_

Round 1

Reviewer 1 Report

The study examines the differences in pedestrian behavior at pedestrian crossings depending on whether they are communicating with autonomous vehicles or conventional vehicles. The rise of autonomous and connected mobility is prompting the need for research on the dynamics and behavior of users in this new form of transportation. Therefore, I consider the subject matter of the study to be current, interesting and with important practical applications in the transport and road safety sector.

Regarding the content, I consider that the article is well thought out. The introduction and methodology are complete, specifying the objectives and main hypotheses of the study. The results are clear, and the discussion adequately contrasts the data obtained with other similar studies in the same field of study.

As for the format, citations should be in numerical format (e.g. [1]) without the names of the authors cited, as specified in the journal format. In addition, references also do not follow the journal's guidelines.

Author Response

As for the format, citations should be in numerical format (e.g. [1]) without the names of the authors cited, as specified in the journal format. In addition, references also do not follow the journal's guidelines.

-> Thank you for reading our manuscript in detail, and pointing out about the format. We carefully revised them.

Reviewer 2 Report

The paper presents a study about differences in pedestrian behaviour at crosswalks when they interact with conventional vehicles or automated vehicles. The study focuses on communication between vehicle and pedestrian and explore its different behaviours regarding the type of vehicle: conventional and automated. The paper is well organized and touches a very interesting topic related with road safety. Its main objective is very relevant to better understand the future interaction between pedestrians and automated vehicles, fostering researchers to develop new strategies and technological solutions to enhance the interaction between these road users in a safe way.

The paper is clear and relatively well organised but has too much sub-sections in the section “2. Materials and Methods”. English is relatively good, but if revised by a native speaker is possible to improve the overall quality of the paper. There are very few typos. 

In terms of content there are several issues that need to be reformulated and clarified.

The authors should include, at the end of the “Introduction” section, a paragraph that summarizes the structure and the contents of all other sections;

One of the most relevant aspects of the study carried out is related to the communication between the vehicle and the pedestrian. From reading the paper, it is not clear whether the drivers of the conventional vehicle were instructed to try to communicate in a similar way. This point is very important because different forms of communication could lead to different pedestrian behaviour and this could introduce a bias in the results and consequently in the conclusions.

The description of the experiment must be detailed. It is relevant to know details about the way how the information was collected and the procedure itself. For example:
 - the position of the three cameras (the inclusion of a scheme with the relative position of the cameras to the crosswalk and roadway );
 - how the data was collected (e.g. positions of vehicles and pedestrians), manually or automatically (and how this was made);
 - the vehicle is stopped at a location 60 m from the crosswalk, this means that it accelerated until reach 20 km/h and the braked to stop near the crosswalk. The pedestrian that is waiting to cross the road see this procedure. Does this not affect his behaviour? Why did you choose this procedure?;
 - each participant only crosses the road once?;
 - Do pedestrians cross alone or in groups? This is not explicitly indicated;
 - how do pedestrians communicate with the vehicle (CV or AV) and vice-versa?;

About the questionnaire and the interview. Only one question has been asked? To ask, “Did you perceive safety?” is enough to know if the participants felt safe during their crossing? What was asked in the interview? Regarding this aspect no results (survey and interview responses) are presented in the section 3 but they are discussed in section 4. The results of the survey and interviews should also be presented in the section 3.

In conclusion the paper needs some improvements. However, by clarifying the aspects that were highlighted, it is possible to have a paper with very interesting and relevant contributions to the study of pedestrian behaviour in crossings when interacting with different types of passenger vehicles.

Author Response

Thank you for reading our manuscript in detail, and pointing out valuable comments. We carefully revised them as below.

Comment: The paper is clear and relatively well organised but has too much sub-sections in the section “2. Materials and Methods”. 

-> We restructured sub-section “2. Materials and Methods” to be more readable.

Comment: English is relatively good, but if revised by a native speaker is possible to improve the overall quality of the paper. There are very few typos. 

-> We revised the English by a native speaker again based on this comment.

Comment: In terms of content there are several issues that need to be reformulated and clarified.

The authors should include, at the end of the “Introduction” section, a paragraph that summarizes the structure and the contents of all other sections;

-> We summarized the structure and the contents of all other sections at the end of the “Introduction” section.

Comment: One of the most relevant aspects of the study carried out is related to the communication between the vehicle and the pedestrian. From reading the paper, it is not clear whether the drivers of the conventional vehicle were instructed to try to communicate in a similar way. This point is very important because different forms of communication could lead to different pedestrian behaviour and this could introduce a bias in the results and consequently in the conclusions.

-> We were afraid that our explanations were not sufficient. The automated vehicle and the conventional vehicle were controlled to be same vehicle behavior by automatic running. Therefore, there were no bias by drivers. We added these explanations.

Comment: The description of the experiment must be detailed. It is relevant to know details about the way how the information was collected and the procedure itself. For example:

 - the position of the three cameras (the inclusion of a scheme with the relative position of the cameras to the crosswalk and roadway );

 - how the data was collected (e.g. positions of vehicles and pedestrians), manually or automatically (and how this was made);

 - the vehicle is stopped at a location 60 m from the crosswalk, this means that it accelerated until reach 20 km/h and the braked to stop near the crosswalk. The pedestrian that is waiting to cross the road see this procedure. Does this not affect his behaviour? Why did you choose this procedure?;

 - each participant only crosses the road once?;

 - Do pedestrians cross alone or in groups? This is not explicitly indicated;

 - how do pedestrians communicate with the vehicle (CV or AV) and vice-versa?;

-> - I added a figure about three camera positions. 

- Data about positions of vehicles and pedestrians were collected by both automatically (radar attached on vehicle) and manually (Observers confirmed the positions by video).

- Pedestrian could not watch the vehicle start, because staff controlled the pedestrian's body direction. Also, staff controlled pedestrian and vehicle positions. Therefore, this method intended to minimize the bias. 

- Each participant crossed the road once.

- After instruction by staff, pedestrians crossed alone.

- Behavior was main communication tool by both pedestrians and vehicles.

We added these descriptions in the manuscript.

Comment: About the questionnaire and the interview. Only one question has been asked? To ask, “Did you perceive safety?” is enough to know if the participants felt safe during their crossing? What was asked in the interview? Regarding this aspect no results (survey and interview responses) are presented in the section 3 but they are discussed in section 4. The results of the survey and interviews should also be presented in the section 3.

-> Several questions have been asked. Regarding the psychological aspect, we not only asked “Did you perceive safety?”, but also “confidence in communication” and “Feeling to be yielded by vehicle”. In statistical test, these questions were highly correlated, therefore, we only used the question “Did you perceive safety?” in our analysis.

Reviewer 3 Report

This paper is well written and filla  gap in the new literature on the interactions between pedestrians and AVs. 

I do not have major comments, except for two:

1. It would be interesting to add more details on the "wizard of Oz". The two papers cited are hard to get (conference proceedings) and the photo is quite puzzling as to how it works.

2. It is not clear to me how these results will be useful to policy makers or other stakeholders responsible for AV and road safety? Can you be more explicit on this in the conclusion? Or at least where we should head after knowing what is exposed in this paper...

Author Response

Thank you for reading our manuscript in detail, and pointing out valuable comments. We carefully revised them as below.

Comment: 1. It would be interesting to add more details on the "wizard of Oz". The two papers cited are hard to get (conference proceedings) and the photo is quite puzzling as to how it works.

-> We added explanation about “wizard of Oz”. The Wizard of Oz experiment method has been used within the field of robot-human interaction for a long period. In this method, the human simulated to test robot behaviors. In the present study, human hid in driver seat, and manipulated vehicle.

Comment: 2. It is not clear to me how these results will be useful to policy makers or other stakeholders responsible for AV and road safety? Can you be more explicit on this in the conclusion? Or at least where we should head after knowing what is exposed in this paper…

-> This research did not directly contribute to policy makers or other stakeholders responsible for AV and road safety, however, will be basic information for several AV’s communication design, such as vehicle behavior and external HMI for communication.